# Infants and Newborns with Atypical Teratoid Rhabdoid Tumors (ATRT) and Extracranial Malignant Rhabdoid Tumors (eMRT) in the EU-RHAB Registry: A Unique and Challenging Population

**DOI:** 10.3390/cancers14092185

**Published:** 2022-04-27

**Authors:** Karolina Nemes, Pascal D. Johann, Mona Steinbügl, Miriam Gruhle, Susanne Bens, Denis Kachanov, Margarita Teleshova, Peter Hauser, Thorsten Simon, Stephan Tippelt, Wolfgang Eberl, Martin Chada, Vicente Santa-Maria Lopez, Lorenz Grigull, Pablo Hernáiz-Driever, Matthias Eyrich, Jane Pears, Till Milde, Harald Reinhard, Alfred Leipold, Marianne van de Wetering, Maria João Gil-da-Costa, Georg Ebetsberger-Dachs, Kornelius Kerl, Andreas Lemmer, Heidrun Boztug, Rhoikos Furtwängler, Uwe Kordes, Christian Vokuhl, Martin Hasselblatt, Brigitte Bison, Thomas Kröncke, Patrick Melchior, Beate Timmermann, Joachim Gerss, Reiner Siebert, Michael C. Frühwald

**Affiliations:** 1Swabian Children’s Cancer Center, Paediatric and Adolescent Medicine, University Medical Center Augsburg, 86156 Augsburg, Germany; karolina.nemes@uk-augsburg.de (K.N.); pascal.johann@uk-augsburg.de (P.D.J.); mona.steinbuegl@uk-augsburg.de (M.S.); miriam.gruhle@uk-augsburg.de (M.G.); 2Division of Pediatric Neurooncology, German Cancer Consortium (DKTK), German Cancer Research Center (DKFZ), 69120 Heidelberg, Germany; 3Institute of Human Genetics, Ulm University, Ulm University Medical Center, 89081 Ulm, Germany; susanne.bens@uni-ulm.de (S.B.); reiner.siebert@uni-ulm.de (R.S.); 4Dmitry Rogachev National Medical Research Center of Pediatric Hematology, Oncology and Immunology, 117997 Moscow, Russia; denis.kachanov@fccho-moscow.ru (D.K.); margarita.teleshova@fccho-moscow.ru (M.T.); 5Velkey László Child’s Health Center, BAZ County Hospital and University Teaching Hospital, 3521 Miskolc, Hungary; drhauserp.gyekig@bazmkorhaz.hu; 6Department of Pediatric Hematology and Oncology, University Children’s Hospital of Cologne, 50924 Cologne, Germany; thorsten.simon@uk-koeln.de; 7Department of Pediatric Hematology and Oncology, Pediatrics III, University Hospital of Essen, 45147 Essen, Germany; stephan.tippelt@uk-essen.de; 8Center for Child and Adolescent Medicine, Department of Hematology and Oncology, Städtisches Klinikum Braunschweig gGmbH, 38118 Braunschweig, Germany; w.eberl@klinikum-braunschweig.de; 9Center of Child and Adolescent Medicine, Department of Pediatric Oncology and Hematology, University Hospital Erlangen, 91054 Erlangen, Germany; martin.chada@uk-erlangen.de; 10Department of Pediatric Oncology, Hospital Sant Joan de Déu Barcelona, 08950 Barcelona, Spain; vsantamaria@sjdhospitalbarcelona.org; 11Department of Pediatric Hematology and Oncology, Children’s Hospital of Hannover, 30625 Hannover, Germany; lorenz.grigull@ukbonn.de; 12Department of Pediatric Oncology and Hematology, Charité–Universitätsmedizin Berlin, Corporate Member of Freie Universität Berlin, Humboldt-Universität zu Berlin, 13353 Berlin, Germany; pablo.hernaiz@charite.de; 13Department of Pediatric Hematology and Oncology, University Würzburg, 97080 Würzburg, Germany; eyrich_m@ukw.de; 14Children’s Health Ireland at Crumlin, D12N512 Dublin, Ireland; jane.pears@olchc.ie; 15Hopp Children’s Cancer Center (KiTZ), 69120 Heidelberg, Germany; till.milde@med.uni-heidelberg.de; 16Clinical Cooperation Unit (CCU) Pediatric Oncology, German Cancer Research Center (DKFZ), 69120 Heidelberg, Germany; 17German Consortium for Translational Cancer Research (DKTK), 69120 Heidelberg, Germany; 18Department of Pediatric Oncology, Hematology and Immunology, Center for Child and Adolescent Medicine, Heidelberg University Hospital, 69120 Heidelberg, Germany; 19Department of Pediatric Hematology and Oncology, Asklepios Hospital Sankt Augustin, 53757 Sankt Augustin, Germany; h.reinhard@asklepios.com; 20Children’s Hospital Karlsruhe, 76133 Karlsruhe, Germany; alfred.leipold@klinikum-karlsruhe.de; 21Princess Máxima Center for Pediatric Oncology, 3584 CS Utrecht, The Netherlands; m.d.vandewetering-4@prinsesmaximacentrum.nl; 22Oncology Department, University Hospital S. João, 4200-319 Porto, Portugal; mjgildacosta@chsj.min-saude.pt; 23Department of Pediatric Hematology and Oncology, Medical University of Kepler, 4040 Linz, Austria; georg.ebetsbergerdachs@kepleruniklinikum.at; 24Department of Pediatric Hematology and Oncology, University Children’s Hospital Münster, 48149 Münster, Germany; kornelius.kerl@ukmuenster.de; 25Pediatric Oncology Center, Helios Klinikum Erfurt, 99089 Erfurt, Germany; andreas.lemmer@helios-gesundheit.de; 26St. Anna Kinderspital and Children’s Cancer Research Institute, Department of Paediatrics, Medical University of Vienna, 1090 Vienna, Austria; heidrun.boztug@stanna.at; 27Department of Pediatric Hematology and Oncology, University of Saarland, 66424 Homburg, Germany; rhoikos.furtwaengler@uniklinikum-saarland.de; 28Department of Pediatric Hematology and Oncology, University Hospital Hamburg-Eppendorf, 20251 Hamburg, Germany; kordes@uke.de; 29Department of Pathology, Section of Pediatric Pathology, University Hospital Bonn, 53127 Bonn, Germany; christian.vokuhl@ukbonn.de; 30Institute of Neuropathology, University Hospital Münster, 48149 Münster, Germany; hasselblatt@uni-muenster.de; 31Department of Diagnostic and Interventional Radiology, University Medical Center Augsburg, 86156 Augsburg, Germany; brigitte.bison@uk-augsburg.de (B.B.); thomas.kroencke@uk-augsburg.de (T.K.); 32Department of Radiation Oncology, University of Saarland, 66424 Homburg, Germany; patrick.melchior@uks.eu; 33Department of Particle Therapy, University Hospital Essen, West German Proton Therapy Centre Essen (WPE), West German Cancer Center (WTZ), German Cancer Consortium (DKTK), 45147 Essen, Germany; beate.timmermann@uk-essen.de; 34Institute of Biostatistics and Clinical Research, University of Münster, 48149 Münster, Germany; joachim.gerss@ukmuenster.de

**Keywords:** *SMARCB1*, atypical teratoid rhabdoid tumors, extracranial malignant rhabdoid tumor, RTPS1, RTPS2, germline mutation, EU-RHAB registry

## Abstract

**Simple Summary:**

Malignant rhabdoid tumors (MRT) are deadly tumors that predominantly affect infants and young children. Even when considering the generally young age of these patients, the treatment of infants below the age of six months represents a particular challenge due to the vulnerability of this patient population. The aim of our retrospective study was to assess the available information on prognostic factors, genetics, toxicity of treatment and long-term outcomes of MRT. We confirmed that, in a cohort of homogenously treated infants with MRT, significant predictors of outcome were female sex, localized stage, absence of a GLM and maintenance therapy, and these significantly favorably influence prognosis. Stratification-based biomarker-driven tailored trials may be a key option to improve survival rates.

**Abstract:**

**Introduction:** Malignant rhabdoid tumors (MRT) predominantly affect infants and young children. Patients below six months of age represent a particularly therapeutically challenging group. Toxicity to developing organ sites limits intensity of treatment. Information on prognostic factors, genetics, toxicity of treatment and long-term outcomes is sparse. **Methods:** Clinical, genetic, and treatment data of 100 patients (aged below 6 months at diagnosis) from 13 European countries were analyzed (2005–2020). Tumors and matching blood samples were examined for *SMARCB1* mutations using FISH, MLPA and Sanger sequencing. DNA methylation subgroups (ATRT-TYR, ATRT-SHH, and ATRT-MYC) were determined using 450 k / 850 k-profiling. **Results:** A total of 45 patients presented with ATRT, 29 with extracranial, extrarenal (eMRT) and 9 with renal rhabdoid tumors (RTK). Seventeen patients demonstrated synchronous tumors (SYN). Metastases (M+) were present in 27% (26/97) at diagnosis. A germline mutation (GLM) was detected in 55% (47/86). DNA methylation subgrouping was available in 50% (31 / 62) with ATRT or SYN; for eMRT, methylation-based subgrouping was not performed. The 5-year overall (OS) and event free survival (EFS) rates were 23.5 ± 4.6% and 19 ± 4.1%, respectively. Male sex (11 ± 5% vs. 35.8 ± 7.4%), M+ stage (6.1 ± 5.4% vs. 36.2 ± 7.4%), presence of SYN (7.1 ± 6.9% vs. 26.6 ± 5.3%) and GLM (7.7 ± 4.2% vs. 45.7 ± 8.6%) were significant prognostic factors for 5-year OS. Molecular subgrouping and survival analyses confirm a previously described survival advantage for ATRT-TYR. In an adjusted multivariate model, clinical factors that favorably influence the prognosis were female sex, localized stage, absence of a GLM and maintenance therapy. **Conclusions:** In this cohort of homogenously treated infants with MRT, significant predictors of outcome were sex, M-stage, GLM and maintenance therapy. We confirm the need to stratify which patient groups benefit from multimodal treatment, and which need novel therapeutic strategies. Biomarker-driven tailored trials may be a key option.

## 1. Introduction

Malignant rhabdoid tumors (MRT) are embryonal tumors with a homogeneous genetic background and an isolated occurrence of aberrations in *SMARCB1* or (rarely) *SMARCA4*. Functionally, these lesions compromise the physiological function of SWI/SNF, a multiprotein complex that acts as an epigenetic remodeling protein. The inactivation of SWI/SNF gives rise to MRT of the CNS (ATRT, atypical teratoid rhabdoid tumor), extracranial within the kidneys (RTK, RT of the kidney) or other soft tissues (eMRT, extracranial extrarenal malignant RT) (e.g., liver, neck, thorax, retroperitoneum, pelvis) [1,2].

MRTs predominantly affect young children with a peak age between 12 and 35 months [3,4,5,6]. Even when considering the generally young age of these patients, the treatment of infants below an age of twelve months represents a particular challenge due to the vulnerability of this patient population. The physiological immaturity of developing organs in children below one year, in particular the brain, limits or even precludes the use of invasive therapeutic measures such as surgery or radiotherapy and imposes a particular challenge for the management of chemotherapy-associated toxicities. While pharmacokinetics of commonly used antibiotics in newborns and early infants are well characterized [7], little is known on short- or long-term side effects of cytotoxic drugs in infants. There is a significant risk for leukoencephalopathy, neurologic and neurocognitive impairments in the vulnerable nervous system of the youngest patients, particularly when using intraventricular MTX. The common late effects, such as endocrinopathies, cardiac dysfunction, osteopathy, peripheral neuropathy and ototoxicity, raise concerns as to whether radiotherapy may be either postponed or replaced by alternative therapeutic means and/or classical chemotherapeutics may be given in combination with novel targeted agents [8,9,10,11,12].

Previous data on the feasibility of intensified chemotherapy regimens such as the EU-RHAB regimen in very young children are on record [13,14]; information on long-term outcome and toxicity of treatment is sparse. In particular, the rates of surviving infants and factors influencing outcome have thus far not been investigated. It remains to be determined whether molecular subgroups of ATRT have a similar impact on patient outcome in infants, as it has been described for patients with ATRT of all ages [4].

Here we reviewed the data of 100 uniformly treated infants below the age of 6 months at diagnosis registered into the EU-RHAB database and analyzed clinical and molecular factors that may influence outcome.

## 2. Materials and Methods

### 2.1. The EU-RHAB Registry

The EU-RHAB registry is a European platform providing guidance and reference diagnostics for patients with MRT of all anatomic locations. Its design, procedures and analyses have been published repeatedly [4,5]. Inclusion criteria for this subgroup analysis were (1) histopathological diagnosis of an MRT, according to WHO criteria confirmed by central pathology review, (2) age below 6 months at diagnosis and (3) informed consent by legal guardians to collect patient-related data. In this subgroup analysis of EU-RHAB, *n* = 48 patients were previously analyzed and published from the whole cohort of EU-RHAB [4,5,13]. EU-RHAB has received continuous approval by the Ethics Committee of the University of Münster (ID 2009-532-f-S, last amendment December 2016).

### 2.2. Consensus Multimodal Therapy

An overview of treatment recommendations, details on drug doses and protocol details are provided in Appendix A.

### 2.3. Diagnostic Measures

The Reference Center for Pathology (for eMRT/RTK–Ivo Leuschner (deceased) and ChV, Kiel, since 2019 ChV, Bonn, for ATRT–MH, Münster, Germany) reviewed all tumor samples according to WHO criteria and routinely included immunohistochemistry for SMARCB1/INI1 and SMARCA4/BRG1 [15]. Radiologic response was evaluated according to criteria of the German National Reference Centre for Radiology (for eMRT/RTK–TK, Augsburg, for ATRT–BB, until 2020 Würzburg, since 2020 Augsburg, Germany).

### 2.4. Toxicity

Toxicity was assessed and reported according to version 3.0 of the Common Terminology Criteria for Adverse Events. Reporting of serious adverse events to the registry office was requested, but not mandatory and not monitored.

### 2.5. Tumor Tissue Collection and Genetic Analyses

All samples were collected at diagnosis. Cytogenetic studies including fluorescence-in situ hybridization (FISH), as well as molecular studies including multiplex ligation-dependent probe amplification (MLPA) and sequencing of *SMARCB1*, were performed at reference centers in Kiel (until 2016) and Ulm (since 2016), Germany (RSi), and Hamburg, Germany (RSch/UK) as previously described on blood samples and tumor tissues [16,17]. DNA for MLPA and sequencing was isolated from FFPE tumor material. DNA methylation subgrouping for ATRT (ATRT-TYR, -SHH, -MYC) was performed at the German Cancer Research Center Genomics and Proteomics Core Facility; protocol details are given elsewhere [4,18].

### 2.6. Statistical Analyses

Univariate analysis of overall (OS) and event-free survival (EFS) for basic items such as gender, age, tumor location, distant metastases, synchronicity, extent of resection, germline mutation (GLM), tumor genetics and DNA methylation subgroups were determined using the log rank test. OS was defined as the time from diagnosis until death of any cause. EFS was defined as time from diagnosis until first progression, relapse, death of any cause or last contact. Time-dependent factors such as high dose chemotherapy (HDCT), radiotherapy (RTx), maintenance therapy (MT), achievement of a complete remission (CR) and relapse or progression were evaluated using Cox regression for time-dependent covariates. Multivariate Cox regression identified independent prognostic factors of OS. *p*-values were regarded significant for *p* ≤ 0.05 without adjustment for multiplicity.

## 3. Results

### 3.1. Patient Characteristics

We analyzed clinical data of 100 patients, registered between June 2005 and May 2020 from 13 European countries (Germany, Austria, Switzerland, Russia, Hungary, The Netherlands, Denmark, Ireland, Sweden, Poland, Slovenia, Portugal and Spain). Male and female patients were equally affected (M:F–51:49). The median age at diagnosis was 3 months (range 0–6 months), however 15 patients presented with congenital rhabdoid tumor. In 45 patients (45%), the tumor was located only in the CNS (ATRT) (most commonly in the cerebellum and hemispheres), in 29% (29/100) extracranial, extrarenal (eMRT) (most commonly in cervical region and in the liver), in 9% (9/100) in the kidneys (RTK). Seventeen patients (17%) demonstrated synchronous, multifocal tumors (AT/RT + eMRT = 11, AT/RT + RTK = 6).

Disease without loco-regional lymph node involvement (M0, LN−) was observed in 50.5% (49/97) of patients. Metastatic stage (M-stage) in *n* = 3 patients was not available. A total of 5% (5/97) were diagnosed with loco-regional lymph node involvement (M0, LN+). Metastases (M+) at diagnosis were present in 27% (26/97) (most frequently CNS, liver, lungs, spleen) and synchronous tumors in 17.5% (17/97). M+ included M1 in ATRT. Four patients developed MRT at age of 0, 1, 2 and 6 months following in vitro fertilization (eMRT = 2, RTK = 1, ATRT + eMRT = 1). All four were females, and one of them (diagnosed at age of 1 month) survived for 126 months after diagnosis in continued CR at last follow-up. Clinical and treatment variables are summarized in Figure 1, Table 1 and Table 2.

### 3.2. Clinical Factors Associated with Outcome

Interestingly, sex demonstrated a significant influence on prognosis. Male patients displayed significantly inferior 5-year OS survival (11 ± 5 %) compared to female patients (35.8 ± 7.4 %) in the uni- and multivariate model (Figure 2A, Table 3). Metastases (M+) at diagnosis were associated with a significantly inferior survival compared to localized disease (M0, LN−, M0, LN+) (Figure 2B). Furthermore, the presence of synchronous tumors correlated with significantly inferior prognosis (5-year OS 7.1 ± 6.9% vs. 26.6 ± 5.3%). In the multivariate analysis, metastases and synchronous tumors were independent prognostic factors for survival (Table 3). Anatomic location of the tumor also seemed to impact on survival; the 5-year OS for extracranial rhabdoid tumors was higher compared to ATRTs, however this did not reach statistical significance (33.4 ± 7.8% vs. 21.9 ± 6.8%).

### 3.3. Germline Mutation of SMARCB1 an Important Risk Factor

Analyses of genetic alterations in *SMARCB1/SMARCA4* were available for 83% (83/100) of patients (tumor and/or blood). A single patient with an ATRT demonstrated loss of *SMARCA4/BRG1*. A total of 60% (76/126 alleles in 63 tumors with complete genetic information) of *SMARCB1* alterations were structural variants (partial or whole gene deletions) and 40% (50/126 alleles) were single nucleotide variants and indels. All alterations were truncating. One patient presented a missense mutation in addition to a nonsense mutation. GLM was observed in 55% (47/86) of patients. Single nucleotide variants were predominant (66%, 31/47). Two out of sixteen patients with synchronous tumors had no detectable GLM; one of them presented in tumor tissue only a homogenous deletion of *SMARCB1*. In another patient, no tumor tissue was available. Presence of a GLM had significant predictive power on survival in univariate testing as well as in a multivariate model. Only 7.7% of patients with a GLM survived longer than 5 years (Figure 2C, Table 3).

### 3.4. Distribution of ATRT Subgroups in This Cohort

DNA methylation profiling was available in 50% (31/62) of cases with ATRT or synchronous tumors with ATRT and we categorized the patients into one of the three described molecular subgroups: 42% (*n* = 13) ATRT-SHH, 42% (*n* = 13) ATRT-TYR and 13% (*n* = 4) ATRT-MYC. For the remaining 31 cases, neither DNA nor FFPE material was available to perform the analysis. For this study, we did not consider methylation-based subgrouping of eMRT.

One patient demonstrated divergent subgroups in samples derived from the infra- and supratentorial compartments of the tumor (ATRT-SHH supra- and ATRT-TYR infratentorial); this case has previously been described and discussed in Thomas et al. [19]. The 5-year OS was superior in the ATRT-TYR group (28.1 ± 13.6% vs. 10.2 ± 9.6% for ATRT-SHH and final level not reached for ATRT-MYC and ATRT-SHH/ATRT-TYR). No further significant correlations were detected between epigenetic and clinical factors.

### 3.5. Extent of Surgery

A gross total resection (GTR) as defined by reference radiologic evaluation was achieved in 32% (32/100) of patients; in 29% (29/100) of patients, only a biopsy was possible. In 3% of patients (3/100), due to extensive disease, no operation was attempted including those with synchronous, multifocal tumors. All of them suffered from congenital rhabdoid tumors and previously diagnosed GLM. Patients with GTR demonstrated higher 5-year OS, however this effect did not reach statistical significance (5-year OS 31 ± 8.5% vs. 19.6 ± 5.4%, *p* = 0.24).

### 3.6. Systemic, Intrathecal and Maintenance Chemotherapy

A total of 93% (93/100) of patients received chemotherapy according to EU-RHAB. In 7% (7/100) no chemotherapy at all was given (palliative intent) as decided by a shared decision-making process between the treating team and legal guardians. A total of 47% (44/93) of patients completed chemotherapy: at least 9 courses of chemotherapy (*n* = 30) or 6 courses of chemotherapy and 1 course of high dose chemotherapy (HDCT) (*n* = 14). After 3 courses of chemotherapy, 50.5 % (47/93) of patients achieved either stable disease (SD) or complete remission (CR). Dose reduction due to toxicity was necessary in 37.6% (35/93) of patients, mostly following VCA courses. However, 63% (22/35) of patients with dose reduction achieved SD or CR (Figure 3). Intraventricular MTX was applied to 50% (31/62), intrathecal therapy to 11% (7/62) of patients with ATRT, or in synchronous tumor. HDCT was applied to 17% (16/93); six of these received additional RTx. A total of 6 out of 10 patients treated without RTx achieved SD or CR after HDCT. Four other patients developed progression while on therapy. VOD was noted in 3/6 patients with early relapse or progression. One of these developed VOD after the first course of HDCT and died; the two other patients remain in CR. From six patients treated with additional RTx, *n* = 3 are currently in CR. Significant improvement in survival for patients treated by intrathecal MTX or HDCT was not seen (Table 3). Maintenance chemotherapy was applied to 19% (18/93) of patients. Maintenance chemotherapy included eight courses of oral trofosfamide/idarubicin (TI) and trofosfamide/etoposide (TE) every three weeks. A total of 14 of 18 patients received the same regimen and 4 patients received cyclophosphamide/vinblastine. Maintenance regimen demonstrated in univariate analyses borderline significance, and in multivariate analysis maintenance therapy was an independent prognostic factor for infants with MRT. The effect of maintenance therapy was evaluated using Cox regression with a time-dependent factor (Table 3).

### 3.7. Radiotherapy

A total of 24% (24/100) patients (ATRT = 9, eMRT = 10, RTK = 2, SYN = 3) received radiotherapy (RTx) at a median age of 12 months (1–38 months) (ATRT–12 months (9–38), eMRT–12 months (5–21), SYN–4 (1–21)). RTx was not delayed until the end of therapy. The median time from diagnosis to RTx was 7.5 months (1–36) (ATRT–8 months (4–36); eMRT/RTK–7 months (2–17), SYN–3 months (1–17)). Median doses of 51.8 Gy (10.8–59.4) were applied to tumor bed; four patients received a boost at 10, 22, 13 and 7 months of age. In fifteen patients, localized disease without loco-regional lymph node involvement (ATRT = 8, eMRT = 6, RTK = 1), and in two patients loco-regional lymph node involvement, was observed (eMRT = 1, RTK = 1). In four patients, distant metastases (ATRT = 1, spinal, eMRT = 3, lung, axilla, femur), and in three patients synchronous tumors (ATRT + Thorax, ATRT + cervix = 2), were present at the time of RTx.

Twelve patients (ATRT = 4, eMRT = 6, RTK = 2) received RTx as a consolidation, five patients in CR. Nine patients received RTx following relapse, in two patients with synchronous tumor to slow down rapid progression and one patient as a rescue measure.

Radiotherapy did not offer a significant survival benefit in our series (Table 3). This conclusion also holds true when investigating eMRT/RTK and ATRT patients separately (*p* = 0.35). However, 75% (9/12) of patients remain in CR (*n* = 8) or SD (*n* = 1) after consolidative radiotherapy. One patient (following in vitro fertilization, diagnosed at age of 1 month) remains in CR following salvage therapy due to relapse at a total of 126 months after diagnosis.

### 3.8. Survival

OS and EFS estimates of the cohort at five years were 23.5 ± 4.6% and 19 ± 4.1%, respectively (Figure 4). For the different localizations, OS and EFS were as follows: ATRT (*n* = 41) 21.9 ± 6.8% and 14.6 ± 5.5%, extracranial MRT (eMRT = 28/RTK = 9) 33.4 ± 7.8% and 28.7 ± 7.4%, synchronous tumor (*n* = 15) 7.1 ± 6.9% and 7.1 ± 6.9%. At the time of analyses, 22 of 100 patients had survived, 18 in complete remission (CR), 1 in stable disease (SD) and 3 in progression (PD). The median follow-up of survivors was 52.5 months (3–124). Interestingly, some patients had survived despite negative prognostic factors: distant M+ at diagnosis (*n* = 3), presence of a GLM (*n* = 3), and without GTR (*n* = 11). Moreover 13 patients survived more than 5 years following diagnosis (ATRT = 5, eMRT = 6, RTK = 2), *n* = 6 thereof after RTx (ATRT = 1, eMRT = 4, RTK = 1). This distribution did not differ from the distribution in the whole EU-RHAB study population.

Patients who had achieved a CR after treatment demonstrated a significant survival advantage (*n* = 34) (*p* = 0.0003). A total of 5 patients achieved CR after surgery, *n* = 25 after additional chemotherapy, and *n* = 4 following RTx. Refractory disease and relapse were very poor prognostic factors. In total, 78 patients suffered from relapses or progressions (*n* = 6 thereof treated with palliative intent). In 53 patients, PD occurred locally only; 13 patients demonstrated combined local relapse and distant metastases. Twelve patients suffered from relapse of distant metastasis with local CR (CNS = 8, lungs = 2, liver = 1, lymph node involvement = 1). Survival after relapse or early progression was, in general, poor. Patients with a lack of response or progression while on chemotherapy (analyzed within 4 months from diagnosis) (*n* = 56) demonstrated a hazard ratio (HR) of 54.7 (95% confidence interval 19.7–151.4, *p* < 0.001) and patients with early relapse (analyzed at 12 months from diagnosis) (*n* = 22) a HR of 64.5 (95% confidence interval 20.7–200.9, *p* < 0.001).

### 3.9. Toxicity of Treatment

Toxicity was noteworthy, but manageable. All patients demonstrated grade 3 or 4 hematologic toxicity at any time during therapy. A total of 17 patients experienced therapy-related SAEs (severe adverse events). Twelve had VOD (venoocclusive disease) as defined by the Seattle criteria (PMID: 28759025), and despite defibrotide therapy, three of them died following this severe event. None of them received RTx in parallel. Interestingly, all ten patients (with available GLM analysis) displayed GLM. In patients with GLM, mutation developed at a significantly higher rate of VOD, compared to patients without GLM (*p* = 0.005). It is noteworthy that these patients were among the youngest patients of the whole cohort (Table 4). With respect to other adverse events, one patient developed encephalomalacia, one cerebral sinus thrombosis, one a shunt failure, one sinus tachycardia and one secondary AML. The latter patient was diagnosed with ATRT (at the age of 3 months) and exhibited a GLM in *SMARCB1* (het del c.825_838del14bp). A total of 23 months from diagnosis, and 11 months after achieving CR, this patient developed a secondary AML and died from the AML.

## 4. Discussion

Even if malignant rhabdoid tumors mostly occur in early childhood, the course of children below six months of age merits special investigation. Recent studies by Reddy et al. [3] and our group [4,5] have analyzed the outcome for children of all ages with MRT but, thus far, subgroup analyses of the youngest patients are sparse.

Our analyses pooled the currently largest cohort of infants diagnosed with MRT at an age below 6 months, treated within the same therapeutic framework. Herein, we report the clinical, molecular and outcome data of *n* = 100 infants. Expectedly, survival of this patient collective was poor (5-year OS: 23.5 ± 4.6%) and even lower than for the whole cohort of patients with intracranial or extracranial MRT, but improved when compared to a cohort of congenital MRT (5-year OS ATRT: 34.7 ± 4.5%, eMRT/RTK: 45.8 ± 5.4%, Congenital: 12.6 ± 5.4%) [4,5,13].

Our results are in line with previous reports indicating metastatic stage at diagnosis as an important factor for survival. Our data demonstrated that patients with localized disease (M0, LN−/M0, LN+) demonstrated significantly better survival compared to patients with synchronous tumors or metastasis (Figure 2B). Moreover, in multivariate analysis, advanced stage was an independent negative prognostic factor for survival (HR: 3.3, *p* = 0.0001) (Table 3).

In accordance with previous reports, we demonstrated a higher percentage of germline mutations (GLM) among patients below six months of age, and an association with a rather poor prognosis [4,5]. Our data indicate that the frequency of germline mutations is approximately 55%. As expected, presence of a GLM is associated with a very poor survival (7.7 ± 4.2% vs. 45.7 ± 8.6%, *p* < 0.0009) (Figure 2C). GLM was a significant prognostic factor for prognosis in uni- and multivariate analysis (HR: 2, 95% confidence interval 1.1–3.6, *p* = 0.02) (Table 3).

Due to the high frequency of germline mutations among infants with MRT below six months of age, we strongly recommend genetic counselling of affected families and surveillance. The majority of individuals diagnosed with a GLM reported to date had de novo pathogenic variants of *SMARCB1*, or gonadal mosaicism. Few of them inherited a pathogenic mutation in *SMARCB1* from one of their (healthy) parents. By contrast, the frequency of hereditary germline SMARCA4 mutations is presumably > 50% [20].

The distribution of methylation subgroups confirmed the previously identified predominance of ATRT-TYR in the youngest patients, and a relative lower incidence of ATRT-MYC, which corresponds to the age distribution of the previously described subgroups [4]. Along a similar line, the survival advantage in ATRT-TYR was confirmed for these young children. The 5-year OS was superior in the ATRT-TYR group, which is also in accordance with previous studies [4] (28.1 ± 13.6% vs. 10.2 ± 9.6% of ATRT-SHH and final level not reached for ATRT-MYC and ATRT-SHH/ATRT-TYR).

Interestingly, clinical prognostic factors identified in other cohorts, such as GTR and RTx, did not indicate any significant survival advantage in this cohort. However, RTx was typically given in patients considered at high risk for local recurrence or for patients already having experienced progression under chemotherapy [5,6,21,22].

GTR, which has been reported previously as an important therapeutic predictor of positive outcome [5], was achieved in 32% (32/100) of patients and did not show a significant survival benefit in our cohort (31.1 ± 8.5 % vs. 19.6 ± 5.4 % (*p* = 0.24). Moreover, GTR is difficult to achieve in patients with extensive disease including synchronous, multifocal tumors (mostly in the context of a GLM) and very young, perhaps prematurely born, patients. Unfortunately, most of these children will be extremely young, potentially born prematurely, and thus rather difficult to operate on at all. Especially in ATRT, a GTR may be associated with an unacceptable burden of morbidity due the potential for severe neurological sequelae. A total of 3% of patients (3/100) did not receive any surgery due to extensive disease, including those with synchronous, multifocal tumors.

Radiotherapy for local control is an important modality for early disease control but remains controversial, especially in the youngest patients. This is particularly true for infants with ATRT in whom neurocognitive and cerebrovascular sequelae may occur long after completion of therapy, but also for eMRT of many anatomic locations (e.g., orbit, liver).

One may hypothesize that a radiotherapy-sparing strategy in these very young children could have led to limited use of radiation in patients in clinically already progressed, and thus difficult, situations. Of the 24 irradiated patients, the majority (14 / 24) received RTx as a consolidative measure, *n* = 10 patients received after a progression. On the other hand, RTx was not found to be significantly beneficial (*p* = 0.35). This conclusion also holds true when investigating eMRT/RTK and ATRT patients separately. Further investigations may help to clarify the role of this treatment modality in very young patients.

Interestingly, maintenance therapy (MT) with eight courses of trofosfamide/idarubicin (TI) and trofosfamide/etoposide (TE) every three weeks demonstrated significant survival advantage. In multivariate analysis, MT was an independent prognostic factor for survival (HR: 0.3, 95% confidence interval 0.14–0.84, *p* = 0.02) (Table 3). Due to the small number of patients, the reasons for the improved survival remain speculative (e.g., higher doses of anthracayclins).

Intensive induction chemotherapy may often achieve respectable responses even in the youngest patients. However, the use of chemotherapy in the first year of life has been associated with a higher risk of VOD than in older children [23]. In particular, Actinomycin-D, in conjunction with abdominal irradiation, as often performed for nephroblastoma, has been identified as a causal agent [24]. EU-RHAB treatment includes doxorubicin and cyclophosphamide, both of which may elicit VOD. Indeed, twelve patients from our cohort developed this serious complication. Using defibrotide, nine patients were successfully treated. Additional studies may help to reveal whether an early intervention or prophylaxis with drugs approved for the treatment of VOD may have beneficial effects.

Aiming to discern clinical and therapeutic factors that may have an influence on the outcome of these patients, we found that female sex, localized stage, and absence of a GLM and maintenance therapy positively influence the prognosis.

According to our data, infants with malignant rhabdoid tumors need to be characterized and stratified clinically and molecularly to ultimately improve outcomes [4,5]. Most patients without risk factors profit from conventional therapies, as we have also seen a group of patients (*n* = 14) who are currently in CR. However, most patients with risk factors, such as M+/SYN, GLM or molecular subgroup ATRT-SHH/MYC, still experience a dismal course and often demonstrate therapy resistance. The question remains how to treat these patients: should these patients be treated experimentally at diagnosis?

Whether the subgroup of young patients may specifically benefit from novel, personalized drugs needs to be evaluated. In particular, the surge of cellular immunotherapies (such as CAR-T cells, [25]) may offer a perspective for this challenging patient group.

Our study suffers from a number of shortcomings, mainly due to the rarity of the disease. Although we present the largest patient cohort of children below 12 months with rhabdoid tumor, absolute patient numbers are still relatively small and may underpower some of the statistical analyses. Another downside of our study is the lack of methylation data for a substantial number of tumors, owing to missing material (e.g., small biopsies only).

It needs to be mentioned that our study represents a subgroup analysis. A part of the patient population described here in detail has previously been included in previous analyses by our group [4,5]. However, a separate analysis of such a high-risk collective has thus far not been reported. This is particularly important for the treating clinician as our data reveal long-term survival even in the youngest patients.

## 5. Conclusions

Overall, management of infants with MRTs remains a major challenge due to age, anatomical and physiologic properties of a developing organism and frequent negative prognostic factors such as synchronicity, metastases and the higher rates of a GLM. Neither GTR nor RTx, as important local modalities for infants with MRT, demonstrated significant effect on survival. However, RTx was given to compensate for early progression in a relevant number of cases in our cohort. We suggest that standardized chemotherapy can be beneficial for a subgroup of patients without high risk factors. Moreover, even in unfavorable situations, a therapeutic attempt is justified and may result in unexpected long-term remission. Our data confirm the need to stratify the patients in clinical trials. More biomarker-driven trials are necessary to address the unmet needs of this patient group, which so far has not been able to benefit extensively from the multimodal treatment currently employed in MRT.

## Figures and Tables

**Figure 1 cancers-14-02185-f001:**
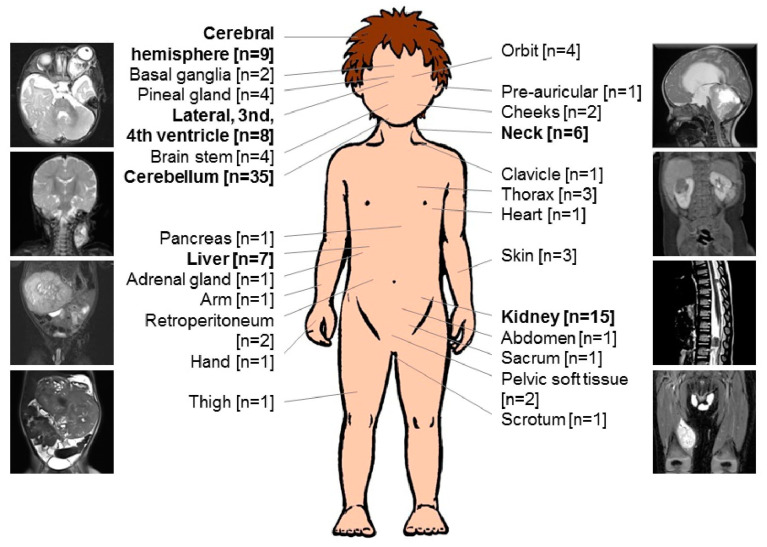
Anatomical localizations of infants with malignant rhabdoid tumor. Anatomical localization of infants with MRT (*n* = 100), included *n* = 17 synchronous tumors, registered between 2005 and 2020 in EU-RHAB database. The most common localization detected in this series, are highlighted in bold.

**Figure 2 cancers-14-02185-f002:**
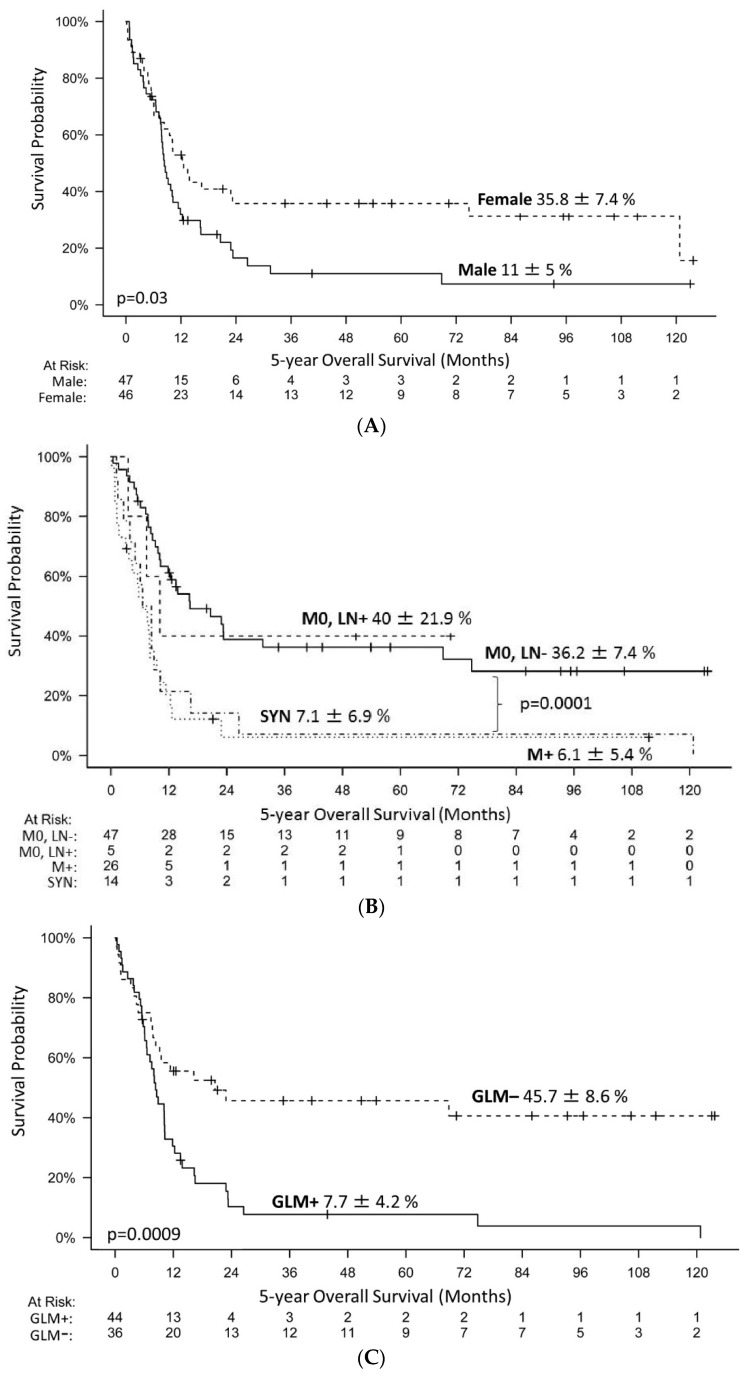
The 5-year overall survival (OS) of 93 consecutive patients treated according to the EU-RHAB consensus therapy. (**A**) The 5-year OS was 35.8 ± 7.4 % for female and 11 ± 5% for male patients. (**B**) The 5-year OS was 36.2 ± 7.4% for patients with localised disease without loco-regional lymph node involvement (M0, LN−), 40 ± 21.9 % for patients with loco-regional lymph node involvement (M0, LN+), 6.1 ± 5.4% for patients with metastasis (M+) at diagnosis and 7.1 ± 6.9% for patients with synchronous tumors (SYN). (**C**) The 5-year OS was 7.7 ± 4.2% for patients diagnosed with germline mutation (GLM+) and 45.7 ± 8.6% for those without (GLM−).

**Figure 3 cancers-14-02185-f003:**
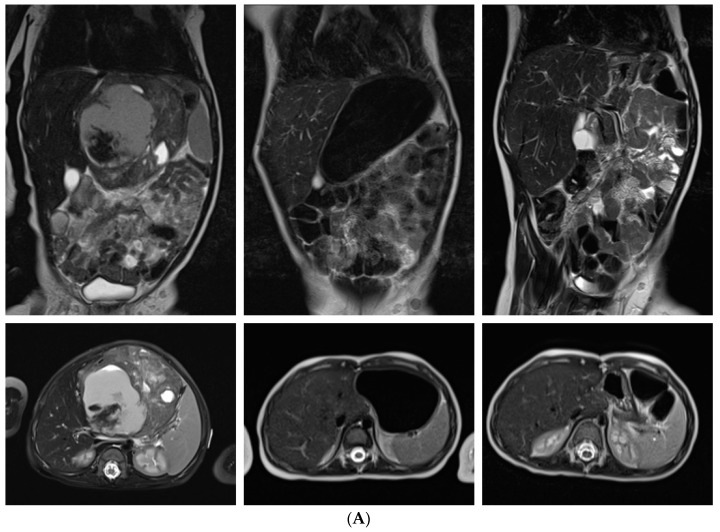
Response to standardized chemotherapy of infants with extensive primary eMRT, imaging results. (**A**) Diagnostic imaging at 3 months of age with inoperable, liver eMRT (8.2 × 8.4 × 7.4 cm), without distant metastasis, without germline mutation. Before GTR (gross total resection), one course ICE (ifosfamide, carboplatin, etoposide) according to EU-RHAB was given. The patient achieved CR (complete remission) and the chemotherapy was continued according to EU-RHAB with four courses of DOX (doxorubicin), three courses of VCA (vincristine, cyclophosphamide, actinomycin) and four courses of ICE. The patient survived 60 months after diagnosis in continuing CR (complete remission). (**B**) Diagnostic imaging at 3 months of age with inoperable, neck eMRT (4.9 × 5 × 6.4 cm), without distant metastasis, without germline mutation. The tumor was resected subtotally (2.5 × 1.2 × 1 cm), and 50% to 25% decrease in tumor volume (IMP–improvement) was detected. After subtotal resection, therapy was continued according to EU-RHAB with two courses of DOX, two courses of ICE, and one course of VCA, and GTR (gross total resection) was performed. After GTR the patient received one course of VCA, high dose chemotherapy (HDCT) and maintenance therapy according to EU-RHAB. The patient achieved CR (complete remission), and one year later RTx was given to tumor bed up to 19.8 Gy with boost up to 16.2 Gy. The patient survived 111 months after diagnosis in continuing CR (complete remission).

**Figure 4 cancers-14-02185-f004:**
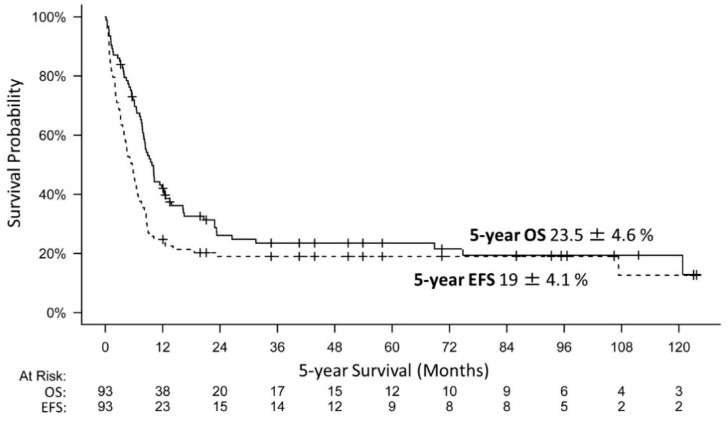
The 5-year overall (OS) and event free survival (EFS) of 93 patients infants with MRT. 7 patients did not receive any chemotherapy, not included in this analysis. The 5-year overall survival (OS) of 93 infants with MRT (ATRT = 41, eMRT = 28, RTK = 9, SYN = 15) was 23.5 ± 4.6%, while the 5-year event free survival (EFS) of the same cohort was 19 ± 4.1%.

**Table 1 cancers-14-02185-t001:** Clinical characteristics of 100 infants with malignant rhabdoid tumors.

	Total
**Median age [months]**	3 (0–6)
**Origin [*n* = 100]**	
Germany/Austria/Switzerland	74
Other countries	26
**Sex [*n* = 100]**	
Female	49
Male	51
**Localization [*n* = 117] ***	
**Intracranial MRT (ATRT)**	62
*Cerebellum*	35
*Cerebral hemisphere*	9
*Lateral, 3*rd, *4*th *ventricle*	8
*Pineal gland*	4
*Brain stem*	4
*Basal ganglia*	2
**Extracranial MRT**	55
*Kidney*	*15*
*Liver*	7
*Neck*	6
*Orbit*	4
*Thorax*	3
*Skin*	3
*Retroperitoneum*	2
*Pelvic soft tissue*	2
*Cheeks*	2
*Pre-auricular*	1
*Clavicle*	1
*Heart*	1
*Pancreas*	1
*Abdomen*	1
*Adrenal gland*	1
*Arm*	1
*Hand*	1
*Thigh*	1
*Sacrum*	1
*Scrotum*	1
**Metastasis [*n* = 97] ****	
M0, LN−	49
M0, LN+	5
M+	26
Synchronous tumor	17

* Anatomical localization of 100 patients, including patients with synchronous tumors (*n* = 17). ** Metastatic stage (M-stage) in *n* = 3 patients not available, M0, LN−; localized disease without loco-regional lymph node involvement, M0, LN+; localized disease with loco-regional lymph node involvement, M+; metastasis.

**Table 2 cancers-14-02185-t002:** Therapy outline and outcome of 100 infants with malignant rhabdoid tumors.

	Total [*n*]
**Gross total resection [*n* = 100]**	
Yes	32
No	65
**Any radiotherapy [*n* = 100]**	
Yes	24
No	76
**High dose chemotherapy [*n* = 93] ***	
Yes	16
No	77
**Maintenance therapy [*n* = 93] ***	
Yes	18
No	75
**Complete remission (of all sites involved) [*n* = 100]**	
Yes	34
*After surgery*	5
*+ chemotherapy*	25
*+ radiotherapy*	4
No	66
**Progression [*n* = 100]**	
No	22
PD on CT **	56
PD after CT ***	22
**SAE [*n* = 17]**	17
VOD	12
Encephalomalacia	1
Sinus vein thrombosis	1
Shunt failure	1
Sinus tachycardia	1
AML	1
**Present status [*n* = 100]**	
Complete remission	18
Stable disease	1
Progressive disease	3
Death	78

* 7 patients did not receive any chemotherapy. ** progression on chemotherapy, analyzed within 4 months from diagnosis, *** progression after chemotherapy, analyzed at 12 months from diagnosis, SAE; serious adverse event, VOD; venoocclusive disease, AML; acute myeloid leukemia.

**Table 3 cancers-14-02185-t003:** Risk factors of overall survival according to univariate and multivariate analyses.

Prognostic Factors	Univariate Analysis *n* = 93	Multivariate Analysis *n* = 81	
	*p*-Value	RR (95% CI)	*p*-Value
**Gender M** versus **F**	**0.03**	2.1 (1.2–3.6)	**0.007**
**M+** versus **M0**	**0.0006**		
**SYN yes** versus **no**	**0.045**		
**M+/SYN** versus **M0, LN−/M0, LN+**		3.3 (1.8–6)	**0.0001**
**GLM yes** versus **no**	**0.0009**	2 (1.1–3.6)	**0.02**
**MYC** versus **TYR ***	**0.0005**		
**GTR yes** versus **no**	0.24		
**HDCT yes** versus **no**	0.23		
**RTx yes** versus **no**	0.35		
**MT yes** versus **no**	0.075	0.3 (0.1–0.8)	**0.02**
**CR yes** versus **no**	**0.0003**		

Gender, localization, synchronous tumors (SYN), metastatic stage (M-stage), metastasis (M+), localized disease with- (M0, LN+) and without loco-regional lymph node involvement (M0, LN−), germline mutation (GLM), 450 k molecular subgroup, gross total resection (GTR), high-dose chemotherapy (HDCT), radiotherapy (RTx), maintenance therapy (MT) were examined. Factors with significance on a univariate and multivariate level are listed. RR; relative risk, CI; confidence interval. * MYC = 3, TYR = 12. The significant *p*-values are highlighted in bold.

**Table 4 cancers-14-02185-t004:** Veno-occlusive disease (VOD).

Patient	Age at Diagnosis (Months)	Primary Tumor Lokalisation	VOD	Current Status(Months)
1	4	Cerebellum + neck (SYN, M0, LN+, GTR cerebellum, GTR neck, CR)	After the third course of VAC, recovered completely	LR (6)–DOD (26)
2	0	Pineal gland + pelvic soft tissue (SYN, M0, GTR pineal gland, not operated pelvis, PD)	After the third course of VAC, recovered completely	PD (2)–DOD (5)
3	4	RTK (M0, GTR, CR)	After the third course of VCA, recovered completely	CR (108)
4	6	Cerebellum (M0, STR, PD)	After the third course of VCA recovered completely	PD (6)–DOD (7)
5	3	Pineal gland + kidney (SYN, M0, LN−, biopsy pineal gland, GTR kidney, PD)	After the third course of VA, recovered completely	PD (6)–DOD (7)
6	0	Cerebellum (M0, STR, SD)	After the third course of VCA, recovered completely	PD (8)–DOD (13)
7	5	Cerebellum (M0, STR, PD)	After the third course of VCA, recovered completely	PD (8)–DOD (11)
8	2	Basal ganglia (M0, biopsy, PD)	After the third course of VCA, recovered completely	PD (5)–DOD (8)
9	4	Cerebellum (M0, PR, SD)	After the first course of HDCT, recovered completely	CR (43)
10	1	Cerebellum (M1, GTR, CR)	After the first course of HDCT	CR (5)–DOT (7)
11	0	Scrotum (M0, LN−, GTR, CR)	After the third course of VAC	CR (1)–DOT (1)
12	1	Heart + Tectum mesencephali (SYN, M0, LN−, biopsy heart, PD)	After achieving one course of VAC	PD (0)–DOT (1)

SYN, synchronous tumor; M0, LN+, localized disease with loco-regional lymph node involvement; M0, LN−, localized disease without loco-regional lymph node involvement; GTR, gross total resection; PR, partial resection; LR, local relapse; CR, complete remission; SD, stable disease; PD, progressive disease; DOD, dead of disease; DOT, dead of treatment; VAC, vincristine, actinomycin D, cyclophosphamide; VCA, vincristine, cyclophosphamide, actinomycin D; VA, vincristine, actinomycin D; HDCT, high-dose chemotherapy.

## Data Availability

The data presented in this study are available on request from the corresponding author.

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
