# Peer review of "Infants and Newborns with Atypical Teratoid Rhabdoid Tumors (ATRT) and Extracranial Malignant Rhabdoid Tumors (eMRT) in the EU-RHAB Registry: A Unique and Challenging Population"

_cancers, 2022, doi:10.3390/cancers14092185_

Round 1

Reviewer 1 Report

Nemes et al provide a comprehensive and interesting overview of the clinical, genetic and therapeutic data of the unique population von newborns and infants (< 6 months) with ATRT and extracranial Malignant Rhabdoid Tumors. Noteably, this paper is well-structured and deserves publication because of the high relevance of the topic. In their text, the authors succeeded in addressing their goal of defining prognostic factors for outcome, genetics, treatment toxicity, and long-term outcome for this population (in comparison to older patients) and in emphasizing the need for new individualized treatment approaches of high-risk patients.

Minor comments:

Spelling mistake: Page 6 (second word: achieved; radiotherapy second passage: consolidation), page 4 last sentence doubled.

Although it’s the currently largest cohort of infants diagnosed with MRT (< 6 months), the number of 100 patients is still relatively small and there is a risk of the underpowerness of statistical effects (as the authors already describe in the discussion). The singular subgroups (tumor localization and genetic mutations/ methylation) cannot be completely compared with each other.

The authors  emphasizing the need for new individualized treatment approaches of high-risk patients and the need of new risk-adapted stratifications, which would be certainly a big goal- however, they did not  mentioned , how they would suggest aoptimal design for a further study- n additional flow chart of a new study and risk stratification design would be very helpful.

Author Response

Reviewer 1:

Comments and Suggestions for Authors

Nemes et al provide a comprehensive and interesting overview of the clinical, genetic and therapeutic data of the unique population in newborns and infants (< 6 months) with ATRT and extracranial Malignant Rhabdoid Tumors. Notably, this paper is well-structured and deserves publication because of the high relevance of the topic. In their text, the authors succeeded in addressing their goal of defining prognostic factors for outcome, genetics, treatment toxicity, and long-term outcome for this population (in comparison to older patients) and in emphasizing the need for new individualized treatment approaches of high-risk patients.

Minor comments:

Spelling mistake: Page 6 (second word: achieved; radiotherapy second passage: consolidation), page 4 last sentence doubled.

We apologize for this mistake. This has now been corrected in the most recent version

Although it’s the currently largest cohort of infants diagnosed with MRT (< 6 months), the number of 100 patients is still relatively small and there is a risk of the underpowerness of statistical effects (as the authors already describe in the discussion). The singular subgroups (tumor localization and genetic mutations/ methylation) cannot be completely compared with each other.

We agree on this well taken point. In our view, this stresses the need for large international series to be collected and analysed e.g. in a metaanalysis. This is already an ongoing project.

The authors  emphasizing the need for new individualized treatment approaches of high-risk patients and the need of new risk-adapted stratifications, which would be certainly a big goal- however, they did not mentioned, how they would suggest an optimal design for a further study- an additional flow chart of a new study and risk stratification design would be very helpful.

We thank the reviewer for raising this important point: The optimal design of trails for HR patients remains somewhat unclear and an international consensus is urgently needed. We suggest that all risk factors should be included in a multivariate, stepwise Cox regression model to create a risk model. In our cohort, only metastases at diagnosis (M+), GLM, and sex remained independent prognostic factors (Table 2). Using this model, we can distinguish two risk groups:

Standard risk: female sex and localized disease with (M0, LN+) or without locoregional lymph node involvement (M0, LN-) and GLM-.

High risk: with one of the features M+ and/or GLM+ and male sex.

We have recently published a risk-adapted stratification for ATRT and eMRT/RTK (Frühwald et al. 2020, Nemes et al. 2021). The independent prognostic factors for ATRTs were age and methylation subgroup, and for eMRT/RTK distant metastases (M+), GTR, and GLM. We determined three subgroups in the ATRT cohort [high risk (<1 year of age + non-TYR); intermediate risk (<1 year of age + ATRT-TYR or ≥1 year of age + non-TYR); standard risk (≥1 year of age + ATRT-TYR)], and two subgroups in the eMRT/RTK cohort (standard risk group: localised disease with- (M0, LN+) and without loco-regional lymph node involvement (M0, LN-), and GTR+ and GLM-; high risk group: presenting with one of the features M+ and/or GTR- and/or GLM+). We have referenced these models in the manuscript.

Reviewer 2 Report

The authors report the EU-RHAB result of 100 young children with AT/RT, eMRT or RTK, including 17 patients with synchronous tumors. Male gender, metastatic or synchronous disease, germline mutation, and lack of maintenance therapy were associated with significant risk of death on multivariant Cox modeling. The study will serve as a benchmark of international multicenter collaboration of MRT studies and the paper is well written. 

Two major questions: 

Would the authors consider AT/RT and eMRT/RTK the same disease or different diseases? Were they treated the same? 
Would some clinical characteristics or treatment arrangements be different when comparing ATRT vs. eMRT/RTK? (e.g. age, stage, GTR rate, RTx rate or timing, High-dose chemotherapy, CR+PR rate, SAE rate) 
For patients with RTK or synchronous tumors after total nephrectomy, would there be some treatment modification/dose adjustments? 
Supplemental Tables - Was intrathecal methotrexate given to all patients including eMRT and TRK without CNS involvement? What was the dose of MTX? Could the authors provide how many doses of intrathecal therapy did this cohort of patients receive (e.g. median and IQR)? 

There are some debates about using Doxorubicin monotherapy in CNS AT/RT. Could the authors provide some rationales about this approach? May the authors compare the EU-RHAB approach to VDC/IE used in Ewing sarcoma? 

Some minor questions/suggestions: 

Was there a reason to choose 6 months of age as an inclusion criteria, but not 3 months or 12 months? 

Line 214 - Does M+ include M1 in AT/RT? Please clarify. 

Maintenance chemotherapy is one of the major predictor of better outcomes in this study. Could the authors provide more details of MT, e.g. regimens, doses, cycles, durations, response/progression? Did they all use the same regimen of TI/TE? Could the survival benefit come from the higher cumulative dose of anthracyclines and etoposide in the MT group? 

Radiotherapy - The authors made thorough analyses and discussions on the timing and use of RTx in this very young population (lines 411-417 and 457-460). Could 1st line, consolidative RT still be beneficial to patients who did not have early progression? (lines 414-415)
What was the median time from diagnosis to RTx? (lines 291-3)
Was all RT delayed to the end of therapy? Could the authors provide the time interval from diagnosis to RTx? (e.g. median and IQR) 
Line 459 - "RTx" with a small x 

Table 2. The case number (n) used to compare each set of variables seem to be different (e.g. MYC vs. TYR may only be available in a subset of AT/RT). Could the authors add in the case number for each line? 
Does the "LN" abbreviation indicate patients without lymph node involvement? If so, a superscript negative mark "-" should be added into the abbreviation to make it more clear (as "LN-"). Other places using this abbreviation may also be revised. 

Table 3. Please remove the "s" from the "Patients" in the first line. 

Figure 2b and line 216 - What does "LR" mean here? Was it described/discussed in the text? 

Line 478 - "achieved" 

Line 485 - Should the "C" in "ICE" be Carboplatin? 

Line 487 - "were" performed. 

Given the relatively high VOD rate in this cohort, would the authors keep the same chemotherapy design for younger infants with ATRT or eMRT/TRK, or would the authors share some different perspectives in their future study designs? 

Author Response

Reviewer 2:

Comments and Suggestions for Authors

The authors report the EU-RHAB result of 100 young children with AT/RT, eMRT or RTK, including 17 patients with synchronous tumors. Male gender, metastatic or synchronous disease, germline mutation, and lack of maintenance therapy were associated with significant risk of death on multivariant Cox modeling. The study will serve as a benchmark of international multicenter collaboration of MRT studies and the paper is well written. 

Two major questions:

Would the authors consider AT/RT and eMRT/RTK the same disease or different diseases?

Malignant rhabdoid tumors (MRT) are embryonal tumors with a homogeneous genetic background and an isolated occurrence of aberrations in SMARCB1 or (rarely) SMARCA4. While both ATRT and extracranial MRT thus share a common genetic background, the epitranscriptomic level of these diseases differs: ATRT-TYR and ATRT-SHH are pure CNS-rhabdoid tumors with little similarities to extracranial tumors. The subgroup ATRT-MYC, however, does display epigenetic similarities to eMRT, maybe suggesting a common cell of origin. However, taken together, the genetic and at least partly epigenetic communalities of eMRT and AT/RT justifiesa joint analysis

Were they treated the same? 

Both cohorts were treated within the same therapeutic framework. When initiating the EU-RHAB cohort in 2005 there were no large international trials available. In order to take a first step towards understanding the therapeutic framework and to define a standard of treatment, all children within EU-RHAB were subjected to the same regimen. Thus, we were able to collect large data sets on uniformly treated patients and to build a framework in which to perform controlled clinical trials at the international or at least European level (Frühwald et al 2020, Neurooncology, Nemes et al 2021, European Journal of Cancer).

Would some clinical characteristics or treatment arrangements be different when comparing ATRT vs. eMRT/RTK? (e.g. age, stage, GTR rate, RTx rate or timing, High-dose chemotherapy, CR+PR rate, SAE rate).

To clarify this question, most important details are summarized in the table below:

ATRT

%

eMRT/RTK

%

Median age (months)

3 (0 – 6)

3 (0 – 6)

Metastasis (M1 included in ATRT) at diagnosis

10/43*

23

16/37*

43

GLM

13/35*

37

20/35*

57

GTR

5/45

11

18/38

47

RTx

9/45

20

12/38

32

HDCT

6/45

13

5/38

13

CR

15/45

33

17/38

45

PD

37/45

82

25/38

66

Still survived

8/45

18

14/38

37

SAE

15/45 (n=6 VOD)

33

3/38 (n=2 VOD)

8

* data available

For patients with RTK or synchronous tumors after total nephrectomy, would there be some treatment modification/dose adjustments? 

Following lengthy discussions with the renal study tumor specialists when writing the recommendations we decided to exchange cisplatin for carboplatin to account for patients who may be left with one kidney following surgery. Doses of cytostatics were to be reduced according to renal function tests.

Supplemental Tables - Was intrathecal methotrexate given to all patients including eMRT and TRK without CNS involvement?

MTX was to be given only in those cases when it was not certain that the tumor had infiltrated the dura from outside the CNS (perimeningeal sites) as a preventive measure.

What was the dose of MTX?

The dose of MTX was depending on age following comparable Medulloblastoma protocols:

MTX Age-dependent dose (applied via rickham reservoir):

Dose in mg

< 2 year

2-3 years

> 3 years

MTX

0,5 mg

1 mg

2 mg

We have added the dose of MTX to Supplement Table 1 and uploaded the new table.

Could the authors provide how many doses of intrathecal therapy did this cohort of patients receive (e.g. median and IQR)?

Only 7 patients received intrathecal therapy, and the median number of cycles administered was 7.5 (1 - 9).

There are some debates about using Doxorubicin monotherapy in CNS AT/RT. Could the authors provide some rationales about this approach?

At the inception of our trial we collected a small series of patients who were treated with DOX upfront in a window setting (ATRT as well as eMRT). As most patients demonstrated responses to DOX as monotherapy we chose to continue with this recommendation. The data is unfortunately unpublished.

May the authors compare the EU-RHAB approach to VDC/IE used in Ewing sarcoma?

Side to side comparison is not possible, although the therapy is very similar in terms of toxicity and effectiveness, however as CNS and kidney tumors (ATRT and RTK) were included as well we chose ICE and VCA.

Some minor questions/suggestions: 

Was there a reason to choose 6 months of age as an inclusion criteria, but not 3 months or 12 months?

The reason we chose an age of 6 months as the inclusion criterion was the convention, and that the patients at the age of 6 months are mostly less than 10 kg and therefore the doses are different.

Line 214 - Does M+ include M1 in AT/RT? Please clarify. 

Yes, M+ includes also patients with M1 in ATRT. We apologize for the less than clear definition. According to the reviewer’s recommendation we clarified it throughout the manuscript.

Maintenance chemotherapy is one of the major predictor of better outcomes in this study. Could the authors provide more details of MT, e.g. regimens, doses, cycles, durations, response/progression?

Maintenance chemotherapy included eight courses of oral trofosfamide/idarubicin (TI) and trofosfamide/etoposide (TE) every three weeks (trofosfamide, 150 mg/m2 per day per os, divided into 2 doses/d from day 1 to 10; etoposide, 50 mg/m2 per day per os, divided into 2 doses/d from day 1 to 10; idarubicine 5 mg/m2 per day per os, once in the morning on days 1, 4, 7 and 10.

A total of 18 patients with MT, n=14 received therapy as consolidation, thereof n=8 remained in CR; in n=6 developed PD at the median of 3.5 months (1 – 118) after starting the MT, and n=4 received MT after PD.

Did they all use the same regimen of TI/TE?

14 of 18 patients received the same regimen, 4 patients received cyclophosphamide/vinblastine.

Could the survival benefit come from the higher cumulative dose of anthracyclines and etoposide in the MT group? 

Since the number of patients received MT is very small and the group is also very heterogeneous, it is very difficult to answer whether the survival benefit stems from the higher cumulative dose of anthracyclines and etoposide. This issue remains speculative.

Radiotherapy - The authors made thorough analyses and discussions on the timing and use of RTx in this very young population (lines 411-417 and 457-460). Could 1st line, consolidative RT still be beneficial to patients who did not have early progression? (lines 414-415)

Our results show that although RTx did not provide a statistically significant survival benefit, patients who received 1st line, consolidative RTx, demonstrated a certain survival benefit, 75% (9/12) of patients remain still in CR (n=8) or SD (n=1) after consolidative radiotherapy.

What was the median time from diagnosis to RTx? (lines 291-3)

The median time from diagnosis to RTx was 7.5 months (1 - 36), compared with 8 months (4 - 36) in the ATRT group, 7 months (2 - 17) in the eMRI/RTK group, and 3 months (1 - 17) in the SYN group.

Was all RT delayed to the end of therapy?

Both (ATRT, eMRT/RTK) received RTx at a median age of 12 months [ATRT 12 months (9 - 38), eMRT 12 months (5 - 21)], RTx was not delayed until the end of therapy.

Could the authors provide the time interval from diagnosis to RTx? (e.g. median and IQR) 

The median time from diagnosis to RTx was 7.5 months (1 - 36), compared with 8 months (4 - 36) in the ATRT group, 7 months (2 - 17) in the eMRI/RTK group, and 3 months (1 - 17) in the SYN group.

Line 459 - "RTx" with a small x

We apologize for this mistake. According to the reviewer’s recommendation we corrected this typo.

Table 2. The case number (n) used to compare each set of variables seem to be different (e.g. MYC vs. TYR may only be available in a subset of AT/RT). Could the authors add in the case number for each line? 

The numbers used depend on the available variables. In the univariate analysis we analysed 93 patients, in the multivariate analysis 81 patients, in the evaluation regarding the methylation groups we included only the ATRT patients or the synchronous tumors with ATRT, therefore the number of patients here is lower. eMRT methylation subgroups were not considered for this analysis.

We have added the number of cases in this group and the total number of patients analysed in the univariate and multivariate analysis to Table 2.

Does the "LN" abbreviation indicate patients without lymph node involvement? If so, a superscript negative mark "-" should be added into the abbreviation to make it more clear (as "LN-"). Other places using this abbreviation may also be revised. 

"LN" abbreviation indicates patients with lymph node involvement. We apologize for this unclarity. According to the reviewer’s recommendation we corrected this it throughout the manuscript.

Table 3. Please remove the "s" from the "Patients" in the first line. 

We have corrected this typo.

Figure 2b and line 216 - What does "LR" mean here? Was it described/discussed in the text? 

We apologize for this ambiguity. LR means LN+ – lymph node involvement, we corrected this error and uploaded a new Figure.

Line 478 - "achieved" 

We have corrected this typo.

Line 485 - Should the "C" in "ICE" be Carboplatin? 

Indeed the C stands for carboplatin. We apologize for this mistake. According to the reviewer’s recommendation we corrected this error.

Line 487 - "were" performed. 

We have corrected this error.

Given the relatively high VOD rate in this cohort, would the authors keep the same chemotherapy design for younger infants with ATRT or eMRT/TRK, or would the authors share some different perspectives in their future study designs? 

The causes for VOD are manifold. For one high doses such as the ones used in HDCT but of course most prominent the use of actinomycin D may lead into VOD. Fortunately, we have seen only limited numbers of VOD, albeit the problem remains significant. As the prognosis of infants is even poorer than in older children we are hesitant as to remove a rather effective component of the multi-agent chemo. In an in vitro study of rhabdoid tumors (Lünenbürger et al 2010) actinomycin D was one of the most active drug and the compound appears to be an essential ingredient in most protocol for embryonal tumors of the kidneys. Furthermore, it appears that with adequate and long lasting in-hospital hydration most cases of VOD may be prevented.
